# Sulpho-Salicylic Acid Grafted to Ferrite Nanoparticles for n-Type Organic Semiconductors

**DOI:** 10.3390/nano10091787

**Published:** 2020-09-09

**Authors:** Cristian Ravariu, Dan Mihaiescu, Alina Morosan, Bogdan Stefan Vasile, Bogdan Purcareanu

**Affiliations:** 1Department of Electronic Devices Circuits & Architectures, Faculty of Electronics Telecommunications and Information Technology, Polytechnic University of Bucharest, S6 060042 Bucharest, Romania; 2Department of Organic Chemistry “Costin Nenitescu”, Faculty of Applied Chemistry and Materials Science, Politehnica University of Bucharest, S1 011061 Bucharest, Romania; dan.mihaiescu@upb.ro (D.M.); alina.morosan@upb.ro (A.M.); 3Department of Engineering of Oxide Materials and Nanomaterials, Faculty of Applied Chemistry and Materials Science, Politehnica University of Bucharest, S1 011061 Bucharest, Romania; bogdan.vasile@upb.ro; 4S.C. Biotehnos S.A., Gorunului Street 3, 075100 Otopeni, Romania; bogdan.purcareanu@upb.ro

**Keywords:** nanocore, organic shell, green semiconductors, n-type film

## Abstract

A disadvantage of the use of pentacene and typical organic materials in electronics is that their precursors are toxic for manufacturers and the environment. To the best of our knowledge, this is the first report of an n-type non-toxic semiconductor for organic transistors that uses sulpho-salicylic acid—a stable, electron-donating compound with reduced toxicity—grafted on a ferrite core–shell and a green synthesis method. The micro-physical characterization indicated a good dispersion stability and homogeneity of the obtained nanofilms using the dip-coating technique. The in-situ electrical characterization was based on a point-contact transistor configuration, and the increase in the drain current as the positive gate voltage increased proved the functionality of the n-type semiconductor.

## 1. Introduction

The latest advances in materials science successfully serve nanoelectronics’ interests, such as flexible electronic devices with elastomeric substrates [1], field-effect transistors attached to a gold electrode sensing pad for deoxyribonucleic acid hybridization [2], carbon-related materials such as diamond [3], or nanocomposites serving as efficient hole-transporting layers for organic solar cells [4]. Additionally, organic materials have been improved over the last 20 years, and now present superior performance than inorganic materials for thin-film transistors (TFTs) [5]. A convenient method for the deposition of organic materials is dip-coating, as reported in 2019 by other authors for metal–organic frameworks [6]. This method works at room temperature and it is suitable for the deposition of 100 nm organic layers. Polymers with small molecules, such as pentacene, are the most widely used organic semiconductors for p-type materials nowadays [7], as well as for n-type materials under special conditions [8]. The precursors of pentacene are polycyclic aromatic hydrocarbons (PAHs), and their toxicity comes from the ability of these PAHs to bind to deoxyribonucleic acid inside cells, thereby producing disruptive effects [9]. Therefore, the green technologies are much sought after for solar cells [10] and other electronic devices [11]. During an international forum in 2018, the future technology of the third generation of organic light-emitting diodes (OLEDs) for display purposes was eco-friendly defined in terms of low power consumption and long lifetime [12]. However, after carrying out a search, zero results were returned for green technologies for n-type organic transistors, except one regarding green solvents [13].

This work was intended to enlarge the gates toward green organic technologies at room temperature, searching for new types of semiconductors with low toxicity and simple molecular organization. In our previous studies, para-aminobenzoic acid was used to construct a p-type green semiconductor [14]. A non-toxic organic compound, acting as an electron donor, is sulpho-salicylic acid (SSA), the chemical structure of which is presented in Figure 1. SSA can be efficiently attached to the external shell of a ferrite (Fe_3_O_4_) nanocore, providing Fe_3_O_4_–SSA nanoparticles using self-assembling techniques [15]. Essentially, an Fe_3_O_4_ nanocore represents an intrinsic semiconductor and SSA is suitable for organic electronic devices due to its molecular conjugation [16]. The self-assembly of SSA onto the external shell of ferrite nanoparticles easily occurs during the synthesis step, yielding core–shell nanoparticles with a good dispersibility in water. To create a demonstrator, we used a low-cost technology to deposit Fe_3_O_4_–SSA onto a compatible insulator on indium tin oxide (ITO)-coated glass. Finally, we tested the n-type characteristics of the Fe_3_O_4_–SSA film using a point-contact transistor. The Fe_3_O_4_–SSA film was contacted by two probes, i.e., the source and drain, and the ITO film was contacted by a third probe, i.e., the gate. This point-contact transistor, also named pseudo-MOS (Metal Oxide Semiconductor) or Ψ-MOSFET (Metal Oxide Semiconductor Field Effect Transistor) transistor, is specifically used for the in-situ electrical characterization of the conduction in thin semiconductors on insulators [17,18], including organic biomaterials [19,20]. The next paragraph presents the synthesis and microphysical characterization of Fe_3_O_4_–SSA.

## 2. Fe_3_O_4_–SSA Synthesis and Characterization

For the synthesis of the Fe_3_O_4_–SSA nanoparticles, the following substances were acquired from Sigma-Aldrich (Redox Lab Supplies Com S.R.L., Bucharest, Romania): Ferric chloride (FeCl_3_), ferrous sulphate heptahydrate (FeSO_4_·7H_2_O), sodium hydroxide (NaOH), and sulpho-salicylic acid (SSA). During all washing steps, high-purity water of 18.2 MΩ·cm was used. The nanoparticles were prepared from primary ferrite nanocore particles (Fe_3_O_4_) by a modified co-precipitation technique, described in detail in [21]. A 2.5 g Fe^2+^/Fe^3+^ stoichiometric mixture from iron sulphate and iron chloride, in 400 mL H_2_O, was used as the magnetite precursor, and co-precipitation was performed using a 7 g NaOH and 3 g SSA in 400 mL H_2_O solution under continuous stirring.

The final aqueous solution with dispersed Fe_3_O_4_–SSA nanoparticles was microphysically characterized by Fourier transform infrared spectroscopy (FT-IR: Thermo Nicolet 6700 spectrometer, Thermo Fisher Scientific, Waltham, MA, USA), dynamic light scattering (DLS: Zetasizer Nano ZS, Malvern Instruments, Malvern, UK), and transmission electron microscopy (TEM: Carl Zeiss, Oberkochen, Germany). FT-IR spectra were recorded using a ZnSe window H-ATR (Horizontal Attenuated Total Reflectance) mounted on a Thermo Nicolet 6700 spectrometer (Figure 2). The FT-IR spectra of the Fe_3_O_4_–SSA nanoparticles presented a characteristic band assigned to the Fe–O stretching at 556 cm^−1^. The existence of an SSA shell was confirmed by the presence of a characteristic absorption band for the asymmetric stretching vibrations of the carboxylate groups at 1556 cm^−1^, the symmetric stretching vibrations of the sulphonate groups at 1023 and 1160 cm^−1^, and the asymmetric stretching vibrations of the sulphonate groups at 1338 cm^−1^.

The skeletal vibration of benzene ring bonds is evidenced by the presence of a characteristic band at 1426 cm^−1^ in the FT-IR spectra. The band around 1094 cm^−1^ is assigned to the symmetric stretching vibration of a C–C bond between the aromatic ring and the carboxyl in the SSA [16].

The dimensional analysis of the Fe_3_O_4_–SSA synthesized nanoparticles indicates good dispersion stability, by the DLS technique, using the Zetasizer Nano ZS (Malvern Instruments, Malvern, UK). According to the DLS results, the Fe_3_O_4_–SSA nanoparticles present an average hydrodynamic diameter of 50.55 nm, good uniformity proved by the obtained polydispersity index value of 0.187, and very good dispersion stability according to a zeta potential of +45.3 mV (Figure 3).

The TEM results indicate an average size of 20 nm of the internal core of the Fe_3_O_4_–SSA aggregate. The organic external shell was destroyed or diminished at higher beam energies [22], but specific techniques were used in order to highlight the SSA shell (Figure 4a).

The X-ray diffraction (XRD) pattern shows the presence of peaks at the 2θ values of 18.27°, 30.18°, 35.42°, 37.13°, 43.22°, 53.62°, 57.18°, and 62.85, corresponding to the (110), (220), (311), (222), (400), (422), (511), and (440) planes; crystallographic orientation is characteristic of the cubic spinel structure of Fe_3_O_4_ magnetic nanoparticles (Figure 4b).

## 3. Experimental Results and Discussion of the OTFT Demonstrator

First, we present the constructed organic TFT (OTFT) as the demonstrator, which was prepared on an ITO-coated glass substrate, purchased from Bruker Daltonics (Bremen, Germany). The next organic layers were deposited by dip-coating, which is a beneficial method that was recently used for metal–organic frameworks (MOFs) [6].

Polystyrene, approximately 200 nm thick, was deposited onto the ITO face by the dip-coating method at a 70 mm/min extraction rate. The upper Fe_3_O_4_–SSA film, approximately 400 nm in thickness, was deposited onto the polystyrene film by dip-coating at a 90 mm/min extraction rate. The organic film thicknesses were estimated by light reflection in the UV-visible spectrum, based on the dip-coating extraction speed determined in previous experiments [21,22].

The main challenge of film deposition is related to the correlation of the dispersibility of a ferrite core in different solvents. We started to test solvents of different polarities, such as water and ethanol, and we finished by selecting acetonitrile as the best candidate. The first issue we experienced was related to the adherence of the subsequent film (Fe_3_O_4_–SSA) to the previously-deposited film (i.e., polystyrene); here, the solution was to use the acetonitrile solvent. The second issue concerned the low dispersibility of the synthesized nanoparticles in the optimal solvents for subsequent deposition. This problem was solved by making specific modifications to the solvent–shell interaction one by one.

In this way, the transistor body, made from the Fe_3_O_4_–SSA film, was separated from the ITO conductor by polystyrene. The schematic structure of the demonstrator OTFT has direct source and drain contacts on the Fe_3_O_4_–SSA film, as well as a gate probe on the ITO film (Figure 5).

A similar characterization device is the pseudo-MOS transistor, which tests the electrical properties of an Si film on oxide [17]. For other organic semiconductors, similar OTFT test devices, which will be compared to our demonstrator, have similar arrangements for their sources, drains, and gates [23,24].

Our source, drain, and gate probes, made from Al, were mechanically-controlled by the Signatone S-725 micropositioner (Signatone Corporation, Gilroy, CA, USA) to ensure ohmic electrical contact [8]. A source–drain distance of 4000 μm was the channel length of our OTFT (Figure 6).

The demonstrator transistor was biased from a double-stabilized power supply HM8012 (Hameg Instruments GmbH, Mainhausen, Germany). The drain currents were recorded by a Keithley6487 pico-ammeter (Keithley Instruments, Inc., Cleveland, OH, USA). The source was grounded, while the drain and gate electrodes were independently biased from HM8012. The source–drain resistances must measure tens of mega-ohms by a Hameg multimeter HM8012 to indicate that accidental penetration of the probes through the Fe_3_O_4_–SSA film has not occurred. Finally, the OTFT device was ready for testing.

First, the transfer characteristics I_D_-V_GS_ were measured for two V_DS_ values, increasing the positive V_GS_ voltage (Figure 7). The transfer characteristics are represented both on a linear scale (Figure 7a), allowing the extraction of the threshold voltage, V_T_, and a vertical logarithmic scale (Figure 7b), allowing the extraction of the I_ON_/I_OFF_ ratio and the sub-threshold slope (SS). The I_ON_ and I_OFF_ currents are usually the maximum and minimum drain currents from the logarithmic scale. The main feature of a field-effect transistor was fulfilled: The drain current varied with the gate voltage (Figure 7a). Similar-shaped transfer characteristics to other OTFTs with a p-type pentacene [23] or an n-type organic [24] film are also visible in Figure 7a,b. We selected these two transistors [23,24] for comparison for a few reasons: (i) Pentacene is a reference material for organic transistors, such as Si for inorganic devices; (ii) pentacene usually presents p-type conduction [7,23,25], but under special conditions, pentacene offers n-type conduction [8,26]; (iii) we compared the transfer characteristics of our test OTFT with those of the p-type pentacene OTFT—obviously, the chosen I_D_-V_GS_ points of p-type pentacene from [23] were inserted on the axis in Figure 7, considering V_GS_ = |V_GS_|; (iv) for further comparisons, we considered BASF material (BASF is a code of a perylene derivative used as an n-type film in [24]) in Figure 7, because it is an n-type material and it is a more recent development.

The classical OTFT with p-type pentacene and an MoO_3_ insertion [23] provides the best fit of the modulus of the current with our measured current. A few parameters were close in value in the modulus: A threshold voltage (V_T_) of 5 V, conventionally extracted as V_T_ = V_GS_| I_D_ = 1%. I_D,max_; a conduction current (I_ON_) of 16 nA (Figure 7a); an I_ON_/I_OFF_ ratio of 500 and a slope (SS) of 5 V/dec (Figure 7b). These values are inferior to those of another OTFT with an n-type organic film [24], presenting a slope (SS) of approximately 600 mV/dec, a lower threshold voltage (V_T_) of 3.1 V, a higher conduction current (I_ON_) of 1 μA, and a higher I_ON_/I_OFF_ ratio of 10^5^ (Figure 7b). The measured parameters from Figure 7 demonstrate an n-type OTFT with the accumulation channel at V_GS_ > 0. The gate current of the proposed OTFT remained under the detection limit of the pico-ammeter for all tests, indicating that the actual OTFTs did not suffer from gate tunneling or probe penetration.

The output characteristics of our OTFT, I_D_-V_DS_, were measured for different V_GS_ values (Figure 8). For comparison, the output characteristics of another n-type OTFT made from pentacene and contacted by Al source and drain electrodes [26], measured under special illumination conditions by other authors, are included in Figure 8.

Figure 8 reveals an approximate saturation voltage, V_dsat_, of +15 V for our OTFT and of +10 V for the n-type OTFT made from pentacene. However, our OTFT achieved better performance in saturation: Higher saturated drain currents acted upon by much lower gate voltages (i.e., V_G_ = 10 V and I_Dsat_ = 17 nA) than the n-type OTFT from [26] (i.e., V_G_ = 80 V and I_Dsat_ = 5 nA). In our OTFT, the mobility of the carriers under saturation can be estimated by the following simple model [27]:(1)$$I_{D,sat}=\frac{W}{2L}\cdot \mu_{n,sat}\cdot C_{PS}{\left(V_{G}-V_{T}\right)}^{2}$$
where W is the channel depth given by the probe’s diameter (500 μm), L is the channel length given by the distance between the source and drain probes (2 mm), I_Dsat_ is the current from the saturation region (~17 nA for V_G_ = 10 V in Figure 8), C_PS_ is the specific capacity of the polystyrene layer (11 F/cm^2^), V_G_ is the gate voltage (10 V), V_T_ is the threshold voltage (5 V), and μ_n,sat_ is the electrons’ mobility when the OTFT works under saturation.

Equation (1) provides the value, μ_n,sat_ = 0.45 cm^2^/Vs for the carriers mobility in our OTFT at V_G_ = 10 V and V_DS_ > V_dsat_. This value is superior than μ_n,sat_ = 0.054cm^2^/Vs, in the n-film OTFT with pentacene, using the same model (1) and constructive data from [26].

## 4. Discussions about the n-Type Characteristics and the Low Toxicity of the Fe_3_O_4_–SSA Film

In this section, the n-type characteristics of the Fe_3_O_4_–SSA compound are firstly discussed as the electron donor. Some specific conduction mechanisms through the SSA molecules from the external shells may arise. For this purpose, an SSA molecule was simulated by HyperChem molecular modeling software, indicating the following electric charge distribution (Figure 9).

Inside the SSA molecule, the local electronic charge density, expressed by a normalized value at the elementary charge, was computed for each atom. The neutrality condition was fulfilled, because the global electrical charge of the SSA molecule was +2.932–2.93 ≅ 0. However, the sulphonate group, SO_3_H, possessed a net negative electronic charge density of −0.298 (Figure 9). This augments the argument to consider the SSA molecule an electron donor, subsequently offering the n-type behavior of the Fe_3_O_4_–SSA film. On the other hand, the Fe_3_O_4_ film or magnetite was indexed as a semiconductor with a Fermi energy of 3.64 eV and band gap energy of 2.2 eV [28]. Hence, the Fe_3_O_4_ core–shell substrate plays a significant role as a matrix for an intrinsic semiconductor.

The second discussion concerns the low toxicity of the Fe_3_O_4_–SSA compound and its precursors during the technological flow. Ferrite has a low toxicity and by degradation, it can generate Fe^+^, O^−^ ions, usually encountered in the human body and the environment.

A quantitative parameter for the evaluation of toxicity is the lethal dose, defined as a given quantity for the studied toxic substance that is administered per kilogram of the body weight of rats, at which a given percentage of the treated test animals die. The median lethal dose, denoted LD_50_, corresponds to a mortality of 50% from the tested animals after inoculation with the substance [29]. Previously, various studies were performed to evaluate the toxicity of SSA [30]. For rats, the LD_50_ for SSA was established at 700 mg/kg [29]. This value indicates much lower toxicity of SSA than that of PAHs, such benzo[k]fluoranthene (LD_50_ = 14 μg/kg) or other PAHs with an LD_50_ below 90 μg/kg [31]. Even in recent studies, a high cytotoxicity of polyphenolic compounds has been revealed [32], while polyphenols are specifically used for organic semiconductors applied in flexible electronics [33]. On the other hand, the precursor of SSA is salicylic acid—a veritable green compound that acts as a plant hormone or vascular drug [22].

## 5. Conclusions

This paper focused on the green synthesis of functional OTFT structures at very low prices. As a result, we demonstrated that an organic transistor with an Fe_3_O_4_–SSA film is operational. Obviously, many functional parameters have to be further optimized in the coming years to surpass the performance of the current OTFTs.

Herein, we investigated Fe_3_O_4_–SSA material as a candidate for green organic transistors. For this purpose, the synthesis of the Fe_3_O_4_–SSA material was based on co-precipitation. The FT-IR spectra confirmed the existence of SSA, while the TEM imaging captured the Fe_3_O_4_–SSA aggregates. The Fe_3_O_4_–SSA nanoparticles had good dispersion stability according to a zeta potential of +45.3 mV.

The point-contact OTFT transistor with an Fe_3_O_4_–SSA film presented an increasing drain current as the positive gate voltage increased, demonstrating the n-type character of the film. This was the main experimental argument for inducing an electron accumulation channel with a positive gate voltage. Compared to other OTFTs, our Fe_3_O_4_–SSA transistor presented a threshold voltage of approximately 5 V and an I_ON_/I_OFF_ ratio of 500, close to the parameters of a classical pentacene OTFT.

## Figures and Tables

**Figure 1 nanomaterials-10-01787-f001:**
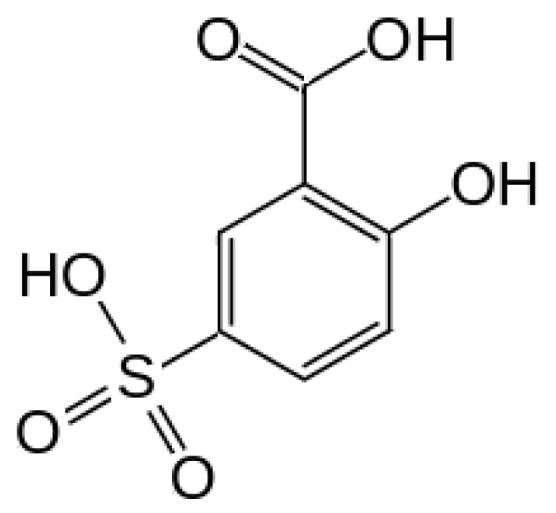
The chemical structure of sulpho-salicylic acid (SSA).

**Figure 2 nanomaterials-10-01787-f002:**
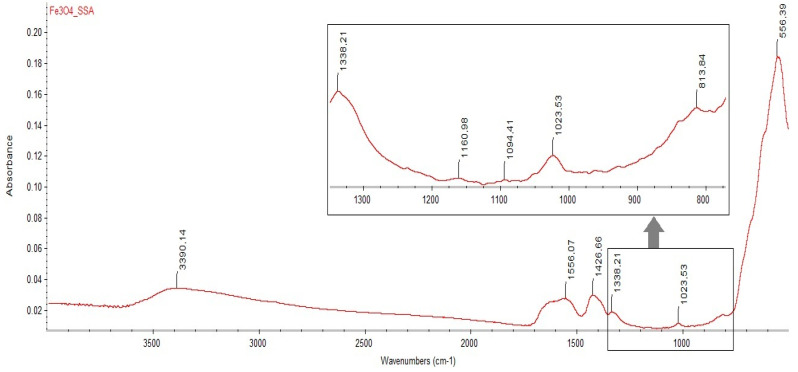
The measured Fourier transform infrared spectroscopy (FT-IR) spectrum of the Fe_3_O_4_–SSA nanoparticles for wavenumbers from 4000 to 500 cm^−1^. The inset provides a detailed overview of wavenumbers between 1338.21 and 813.84 cm^−1^.

**Figure 3 nanomaterials-10-01787-f003:**
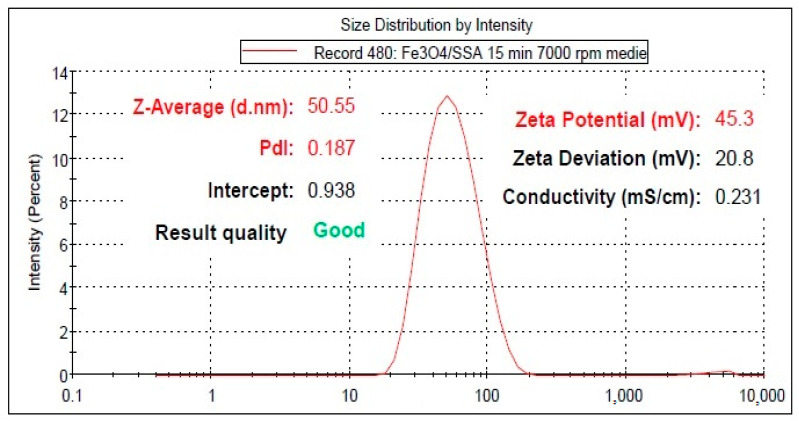
The results of the dynamic light scattering (DLS) analysis for the Fe_3_O_4_–SSA synthesized nanoparticles indicate good dispersion stability.

**Figure 4 nanomaterials-10-01787-f004:**
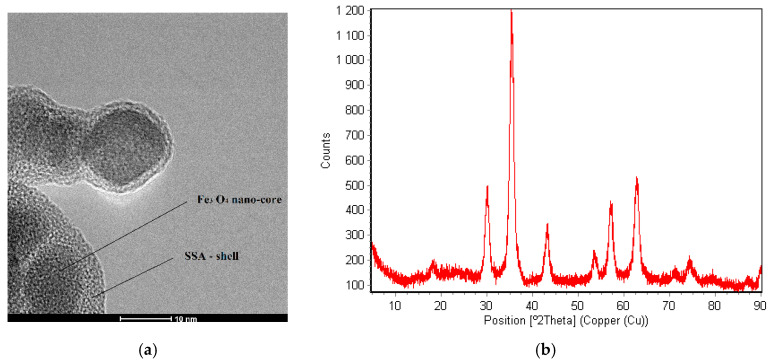
(**a**) The transmission electron microscopy (TEM) image of the synthesized Fe_3_O_4_ nanoparticles, to which the SSA molecules were attached as external shells; (**b**) X-ray diffraction (XRD) pattern of the nanofilm of the Fe_3_O_4_–SSA nanoparticles deposed by dip-coating.

**Figure 5 nanomaterials-10-01787-f005:**
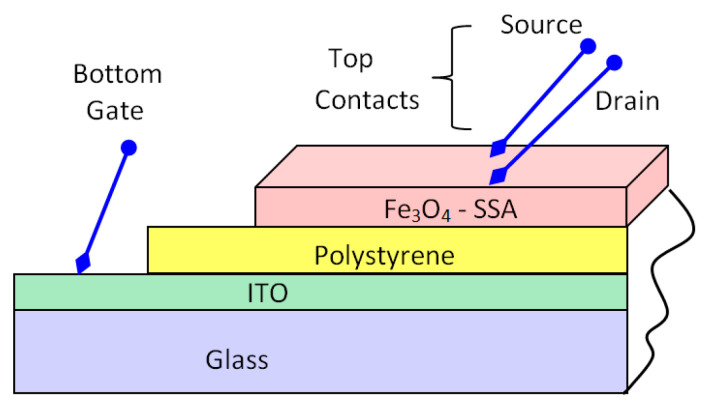
A schematic of the organic thin-film transistor (OTFT) structure, showing the successive layers and the three electrodes places: Gate in contact with the indium tin oxide (ITO), and the source and drain in contact with the Fe_3_O_4_–SSA film.

**Figure 6 nanomaterials-10-01787-f006:**
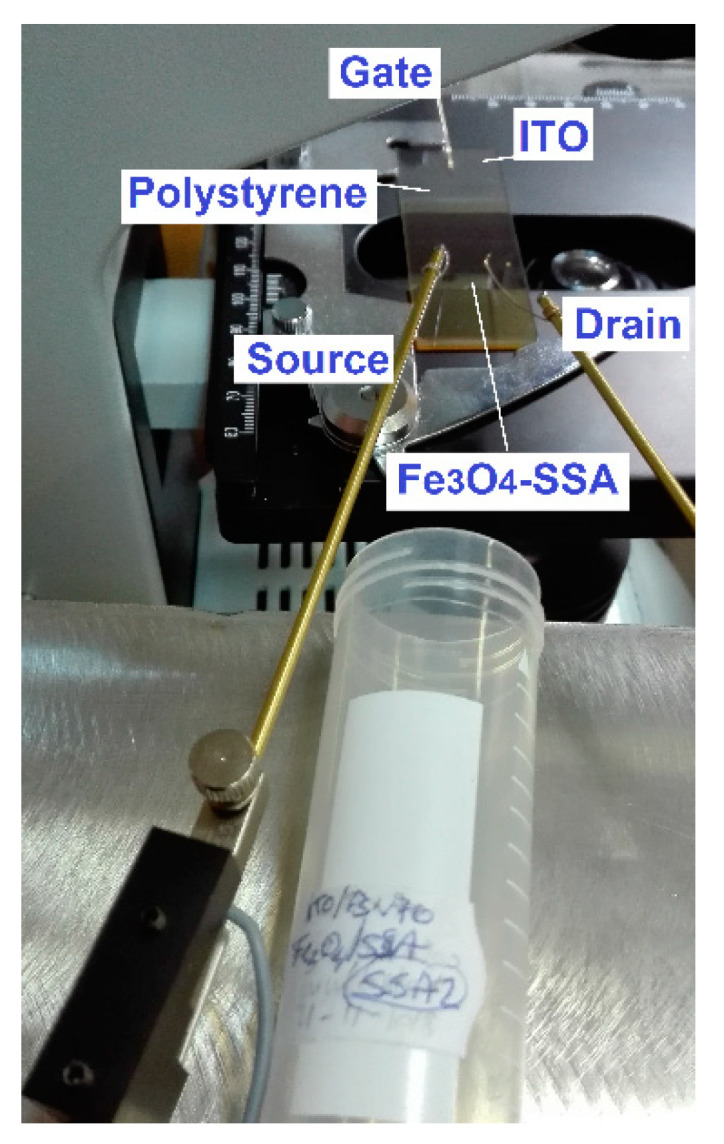
The experimental demonstrator: The source, drain in contact with te Fe_3_O_4_–SSA film (beige-colored), and gate connection to the ITO film. The colorless polystyrene film is longer than the Fe_3_O_4_–SSA film and is shorter than the ITO film, in agreement with Figure 5.

**Figure 7 nanomaterials-10-01787-f007:**
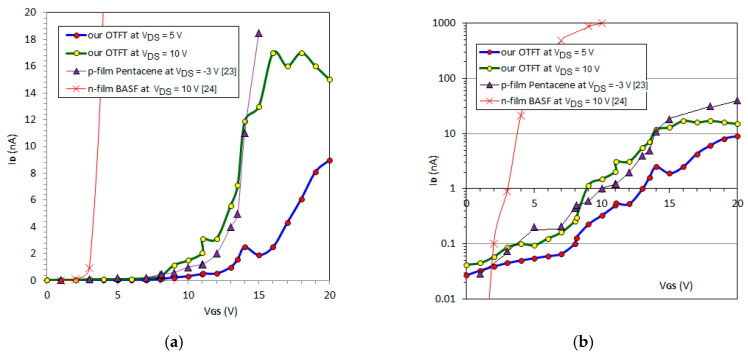
The recorded I_D_-V_GS_ curves: (**a**) Linear scale on both axes; (**b**) log scale on the vertical axis, as well as the chosen experimental points from the literature [23,24].

**Figure 8 nanomaterials-10-01787-f008:**
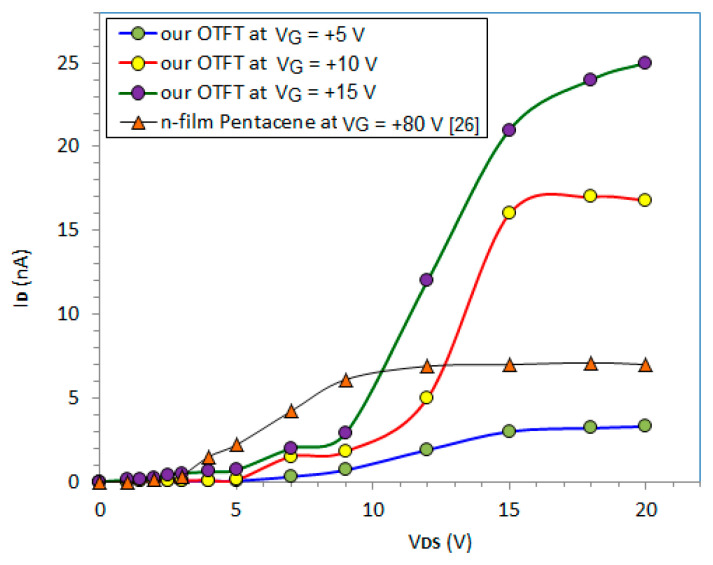
The output characteristics, I_D_-V_DS_, of our OTFT and those of an experimental n-type OTFT made from pentacene [26].

**Figure 9 nanomaterials-10-01787-f009:**
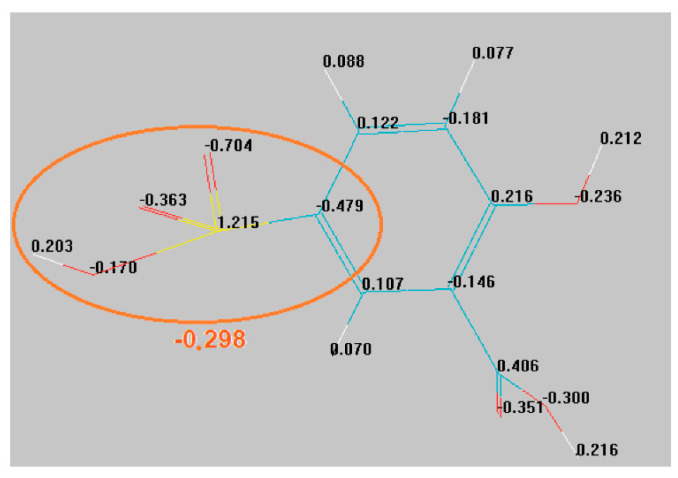
The simulation results of the electrical charge distribution inside the SSA molecule.

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
