# Peer review of "Sulpho-Salicylic Acid Grafted to Ferrite Nanoparticles for n-Type Organic Semiconductors"

_nanomaterials, 2020, doi:10.3390/nano10091787_

Round 1

Reviewer 1 Report

The manuscript has been revised based on the comments, point by point.

I recommend that this paper is suitable to publish Nanomaterials jouranl.

Reviewer 2 Report

The revised version and the response from the authors have addressed the referee's concern. The referee recommends its publication.

This manuscript is a resubmission of an earlier submission. The following is a list of the peer review reports and author responses from that submission.

Round 1

Reviewer 1 Report

This manuscript likes to develop non-toxic semiconductor based on Sulfo-Salicylic acid/Fe3O4 hybrids for organic transistor. However, the major concern is how to define the toxicity of the organic semiconductor? How to compare the toxicity of current hybrid with the state-of-art semiconductors? What is the key significance of developing such hybrid-based transistor with such low performance? These concerns hindered the referee to recommend its publication at its current stage.

Reviewer 2 Report

This manuscript contains the synthesis of an n-type non-toxic semiconductor for organic transistors, using 5-sulfo-salicylic acid grafted on ferrite core-shell and micro-physical characterization by using FTIR, DLS, TEM.The authors have succeeded to achieve that the presently developed Fe3O4-SSA transistor presented similar threshold voltage closer to a pentacene OTFT. The background as well as the results are well presented, and the authors take great care only to limit the development of green synthetic routes and a stable compound, donor of electrons, having a reduced toxicity of n-type semiconductor different from those of pentacene OTFT. Thus, this manuscript could be interesting for the readers in this field of material sciences as well as device fabrication procedures.

Although this paper is not easy to read for general readers due to many abbreviations, this could well serve as the basis for a devel­opment of these fields.

Minor comments

  1. The authors should draw the chemical structures of the compound such as 5-sulfo-salicylic acid used in the present work. And since there are many abbreviations in the text, it is helpful for general readers to explain the origin of these abbreviations.
  2. In Figures in the text the authors should describe the more detailed supplements.
  3. The authors should indicate the corresponding author.
  4. The authors should improve the typos, hyphenation and capital letters as well as

the numbering of the references in the text.

     I recommend this manuscript for publication in Nanomaterials after minor revisions described above.

Reviewer 3 Report

In this manuscript, Cristian Ravariu et al. introduced study to synthesize sulfo-salicylic acid 17 grafted on ferrite core-shell by green synthesis routes and investigate OTFT properties. This study is unique compared to other OTFT through green compound. However a few issues still need to be clarified prior to the further consideration of this manuscript with follow major issue.

1. I think the title should be changed concisely and clearly.
2. This manuscript has to be checked by the professional English editing before submit the revision.
3. “FT-IR”, “DLS”, “TEM” should be written as full name when they are mentioned the first time in the article in line 83 page 2.
4. In XRD data, author should explain and discuss what each peaks of XRD data mean specially.
5. What does mean “An OTFT from 2014” and “an OTFT from 2017”
6. How do you measure thickness and surface of organic layer by dip-coating method? Author should state how to measure thickness/morphology and instrument model in detail.

Reviewer 4 Report

The authors fabricated Sulfo-Salicylic Acid modified ferric oxide core-shell nanoparticles for active layer of OTFT device. I do not think the quality of this manuscript can meet the requirement of Nanomaterials, especially on the characterization. I summary all the comments as follows. 

(1) I doubt the most important electrical characterization of OTFT device. I do not find the transfer characterization (Id vs Vd) and there are two curves in the output characterization. There is no basic saturation region. The authors do not report any field effect mobility data. How to calculate the threshold voltage without the transfer curve?

(2) The authors claim the core-shell nanoparticles but I do not find the clear structures in the TEM images. 

(3) The authors need to mention the crystal orientation in XRD peaks. 

(4)  I can not understand why the authors include the p-type pentacene (Fig. 6) for comparison in this n-type organic semiconductors system.